# Selection pressure on the rhizosphere microbiome can alter nitrogen use efficiency and seed yield in *Brassica rapa*

Joshua Garcia[1], Maria Gannett [1], LiPing Wei[1], Liang Cheng [1], Shengyuan Hu[2], Jed Sparks[3], James Giovannoni [4] & Jenny Kao-Kniffin [1✉]

Microbial experimental systems provide a platform to observe how networks of groups emerge to impact plant development. We applied selection pressure for microbiome enhancement of *Brassica rapa* biomass to examine adaptive bacterial group dynamics under soil nitrogen limitation. In the 9th and final generation of the experiment, selection pressure enhanced *B. rapa* seed yield and nitrogen use efficiency compared to our control treatment, with no effect between the random selection and control treatments. Aboveground biomass increased for both the high biomass selection and random selection plants. Soil bacterial diversity declined under high *B. rapa* biomass selection, suggesting a possible ecological filtering mechanism to remove bacterial taxa. Distinct sub-groups of interactions emerged among bacterial phyla such as *Proteobacteria* and *Bacteroidetes* in response to selection. Extended Local Similarity Analysis and NetShift indicated greater connectivity of the bacterial community, with more edges, shorter path lengths, and altered modularity through the course of selection for enhanced plant biomass. In contrast, bacterial communities under random selection and no selection showed less complex interaction profiles of bacterial taxa. These results suggest that group-level bacterial interactions could be modified to collectively shift microbiome functions impacting the growth of the host plant under soil nitrogen limitation.

[1] School of Integrative Plant Science, Cornell University, Ithaca, NY, USA. [2] Machine Learning Department, Carnegie Mellon University, Pittsburgh, PA, USA. [3] Department of Ecology and Evolutionary Biology, Cornell University, Ithaca, NY, USA. [4] USDA-ARS and Boyce Thompson Institute, Ithaca, NY, USA. ✉email: jtk57@cornell.edu

In recent years, numerous plant microbiome studies have highlighted the intricate links between a plant host and its associated microbiomes[1–3]. In the rhizosphere, microbiomes play a key role in processes that affect plant fitness, such as nutrient cycling[4,5] and disease suppression[6,7]. Given their influence on plant growth and health, microbial consortia in the rhizosphere have become targets for altering plant traits that may help improve agricultural productivity[8,9]. Several studies have demonstrated the ability to alter host traits via selection (i.e. directed microbiome manipulation) for microbial consortia in the root zone. For example, experimental evolution work by Swenson et al.[10] demonstrated that plant biomass could be enhanced in *Arabidopsis thaliana* following inoculation with rhizosphere microbiomes selected based on their association with increased host biomass production over multiple generations. A similar study demonstrated microbiome modification of flowering time in *A. thaliana*[11]. Considering these findings, the creation of robust plant growth promoting rhizosphere microbiomes could be possible through the careful conveyance of group-level selective processes (i.e. selective processes that affect whole groups of organisms rather than individuals alone) for rhizosphere microbial consortia associated with enhanced productivity and fitness.

Microbiome sequencing data can provide a window into the changing communities and dynamics of group-level processes such as nutrient cycling in microbiome experiments over time. Microbial experimental systems that entail simulations of both microbe-microbe and plant-microbe interactions can be designed as empirical studies to track the outcome of selective pressure on a group-level trait[12]. Resulting sequence data of the microbiome can be used to observe the emergence of interaction networks that could signify group-level processes developing through time. The significance of this network profile is the potential record of group selection processes unfolding within a system[13,14]. A key tenet of group selection is that selective pressures at the group- and individual levels constantly interact with one another to produce the observed phenotype of a population[15]. In the context of host-microbe interactions, the microbiome can be shaped heavily by plants when the microbial environment and resource pool are largely defined by the host[16]. The resulting microbiome, in turn, alters different aspects of host growth and development[4]. A better understanding of the plant-microbiome feedback processes in the rhizosphere could reveal how group-level interactions collectively shape host phenotypes.

For microbial experimental systems involving time series sequence data, it is important to consider what types of microbiome interactions can be interpreted in a network profile. Microbial taxa that are positively associated could indicate cooperative or antagonistic behaviors. For example, resource sharing could be extremely costly for individuals given that members in the community may utilize the resource in an uncooperative manner[17], thereby increasing their own fitness while avoiding the cost of cooperation. Network models may not distinguish between these forms of interactions unless the experimental design explicitly controls for contrasting scenarios that determine if more cooperative communities can outcompete others when certain group-level selective pressures are acting on the system[18–20]. In the case of the rhizosphere and its root-associated microbiome, group-level selection could be ubiquitous[4,21,22] and may help explain the high rates of carbon losses from the plant via root exudation[23] as an adaptive strategy to harbor beneficial microbiota in the root zone. Given that these forms of selective pressures are known to influence the microbiome, directed evolution in the rhizosphere for microbial communities that confer an ecosystem-level trait, such as enhanced aboveground biomass production, could be a possibility.

Here, we examined if plant productivity (i.e. aboveground biomass and seed yield) could be enhanced in the non-mycorrhizal host *Brassica rapa* through repeated selection for rhizosphere microbiota associated with increased aboveground biomass production over nine generations of plantings. The objective of this experiment is to examine changes in rhizosphere bacterial group dynamics across generations of selection for high biomass growth of a nonmycorrhizal plant under soil nitrogen limiting conditions. Arbuscular mycorrhizal fungi form associations with most terrestrial plant species that support nitrogen and phosphorus uptake of the plant host[24,25], but the Brassica lineage of plants is dominated by species that do not serve as mycorrhizal hosts[26]. The loss of key symbiosis genes in non-host Brassicas occurred early in the lineage of the Brassicales order[27], which suggests that these plants have evolved different strategies to acquire nitrogen and phosphorus from soil. We hypothesized that selection pressure on rhizosphere microbiomes for increased biomass of a non-mycorrhizal plant host growing under nitrogen-limiting conditions will lead to the assembly of distinct, highly connected and interactive rhizosphere bacterial groups associated with plant host nitrogen uptake strategies. Additionally, we hypothesize that enhanced soil N cycling would be a key mechanism by which the rhizosphere microbiome of a non-mycorrhizal plant host would enhance biomass growth and seed yield.

The selection treatment for high biomass plants was compared with two types of control treatments: plants that were chosen through random selection (random treatment) and plants that were grown with cryopreserved soil microbial inocula that served as controls against microbial adaptation (control treatment). All three selection treatments in the experiment received the same bacterial inoculant in the first generation, derived from organic farm soils. Thus, the initial bacterial community was the same for all three selection treatments but diverged over time across generations of the selection experiment. The microbial inocula consisted of rhizosphere soil dispersed with sterile water into autoclaved soils. Each selection treatment consisted of 15 replicated pots with identical autoclaved soil and *B. rapa* seeds across treatments and generations of plantings, which ensured that the microbial inocula were the dominant variable. The experimental design optimized soil bacterial passage across selection generations through use of the liquid dispersal medium and rapid generation times of the host plant (10-day growth of *B. rapa* Wisconsin Fast Plants). For a detailed illustration of the experimental design, see Fig. 1. We used amplicon sequencing of 16 S rRNA genes to observe shifts in bacterial community composition and interaction profiles across taxa. For bacterial community function, we assessed changes in plant nutrient use efficiency by measuring isotopic nitrogen ($\delta^{15}$N) of plant tissue to indicate potential changes in soil N processing and nutrient metabolism within plants, which is a major determinant of plant productivity and yield.

Our data show *B. rapa* plants in the high biomass treatment had significantly greater seed yield and nitrogen agronomic use efficiency (NUE) compared to plants in the control treatment after the 9th and final generation of selection, with no effect between the random selection and control treatments. Microbial sequencing data from *B. rapa* rhizospheres revealed distinct community profiles between the three selection treatments over the course of the experiment, suggesting distinct rhizosphere microbial communities formed in each treatment. Network analysis using our microbial sequence data showed unique microbial interaction networks between the three treatments, with overall greater connectivity of microbiomes in the high biomass selection treatment. Meanwhile, bacterial communities under random selection and no selection showed less complex

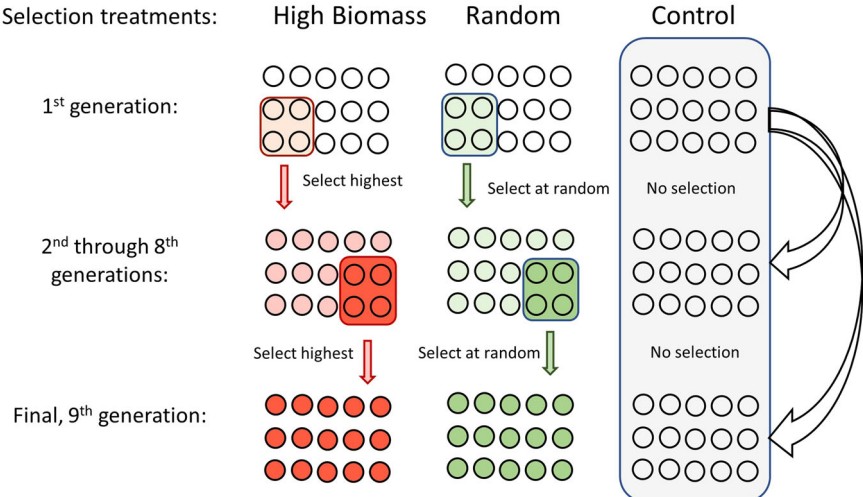

**Fig. 1 Experimental design of selection for rhizosphere microbiomes that enhance plant host yield.** Selection pressure is placed on the rhizosphere microbiome for phenotypic changes (enhanced yield) of the host plant. For the 1st through 8th generations of the high biomass selection treatment, the rhizosphere soil from the four highest yielding units (from $n = 15$ replicates) is composited into microbial inoculants for the subsequent generation of plantings. For the random selection treatment, four units (from $n = 15$ replicates) are selected at random with rhizosphere harvested and used as microbial inoculants for the next planting. The control treatment consists of cryopreserved microbial inoculants that are identical to the start of the 1st generation of plantings and represent non-adaptive microbiomes ($n = 15$ replicates). For all three selection treatments, seeds were collected to determine seed yield and nitrogen use efficiency (NUE) in the final (9th) generation. Units in all treatments and generations of plantings were comprised of the same *B. rapa* seed pool to minimize genetic diversity of the plant host. All units of the three selection treatments in the 1st generation received identical microbial inoculants derived from a mixture of organic farm soils.

interaction profiles of bacterial taxa. Altogether, this experiment shows group-level bacterial interactions could be modified to collectively shift microbiome functions impacting the growth of the host plant under selection pressure.

## Results

**Plant, soil, and microbiome samples**. In the 9th and final generation of the experiment, the plants were allowed to set seed to determine selection pressure outcomes on *B. rapa* seed yield and nitrogen use efficiency. Only the harvested aboveground biomass is reported for plant data from the previous 1st through 8th generations, followed by seed yield in the 9th generation. Likewise, rhizosphere bacteria data are shown for the 1st through 8th generations, and the 9th generation sampling was avoided due to the prolonged growth of seed-bearing plants and low moisture soil conditions (and eventual desiccation) required for seed collection.

**Aboveground dry biomass production and seed yield**. To assess if our selection process enhanced plant productivity in the high biomass selection treatment, we compared the total dry aboveground biomass production and seed yield of the three selection treatments in the 9th generation. Analysis of aboveground dry biomass production (g) revealed significant effects of selection treatment (df = 2, F value = 13.3, $p = 3.28e\text{-}05$). Pot units in the high biomass and random selection treatments had 36% and 38% greater *B. rapa* aboveground dry biomass production compared to pots in the control treatment at the $p = 0.0002$ and $p = 0.0001$ significance levels, respectively (Fig. 2a). Aboveground dry biomass production between the high biomass and random selection treatments did not differ ($p = 0.97$). Analysis of seed yield revealed significant effects of selection treatment (df = 2, F value = 3.7, $p = 0.032$). The seed yield, measured as the total g of seeds produced per pot unit, in the high biomass selection treatment was 62% higher compared to the control treatment at the $p = 0.025$ significance level (Fig. 2b). In contrast, the seed

yield of the random selection treatment did not differ from the high biomass selection treatment ($p = 0.51$) or the control ($p = 0.25$).

**Analysis of plant tissue N content**. Plant N uptake is one of the largest determinants of productivity[28]. We expected that increased soil N cycling would be a key mechanism by which the rhizosphere microbiome of a non-mycorrhizal plant host would enhance biomass growth and seed yield. Thus, we analyzed foliar N data from the 9th generation of the experiment to determine how selection may have altered microbial nutrient cycling activity. Analysis of total foliar N revealed effects of selection treatment in the 9th generation of the experiment at the $p = 0.031$ significance level ($n = 4$, df = 2, F value = 5.2) (Table 1). Plants in the high biomass selection treatment had 60% more total foliar N compared to plants in the control treatment at the $p = 0.028$ significance level, while the total foliar N content of plants in the random selection treatment did not differ from the high biomass ($p = 0.6$) or control ($p = 0.12$) treatments. Analysis of foliar $\delta^{15}N$ values in the 9th generation revealed further effects of selection treatment (df = 2, F value = 15.0, $p = 0.0013$). Plants in the high biomass selection treatment had 43% lower foliar $\delta^{15}N$ values compared to plants in the random selection treatment, and 49% lower foliar $\delta^{15}N$ values compared to plants in the control treatment at the $p = 0.0077$ and $p = 0.0014$ significance levels, respectively (Table 1). The foliar $\delta^{15}N$ values of the random selection and control treatments did not differ from one another ($p = 0.46$). Foliar % N did not differ between any selection treatments (df = 2, F value = 1.7, $p = 0.22$). Selection treatment also had a weak effect on N agronomic use efficiency (NUE), defined as seed yield (g) g$^{-1}$ nitrogen added[28], at a $p = 0.08$ significance level ($n = 4$, df = 2, F value = 3.3). Plants in the high biomass selection treatment had a similarly weak relationship exhibiting a 112% greater NUE compared to the control treatment in the 9th generation ($p = 0.071$) (Table 1). The NUE of the random selection treatment did not differ significantly from the high biomass ($p = 0.65$) and control ($p = 0.27$) selection

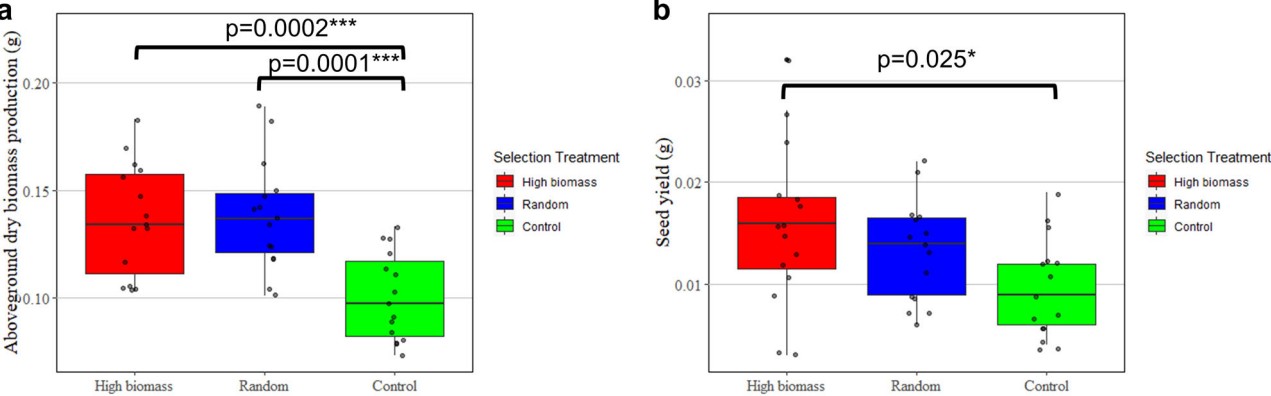

**Fig. 2 Plant productivity at the 9th generation for the high biomass, random, and control selection treatments. a** Box plot displaying the average aboveground biomass production (g) of the pots in each selection treatment after the 9th generation of selection. **b** Box plot displaying the average seed yield (g) of the pots for each selection treatment after the 9th generation of selection. The different selection treatments are represented in color with the high biomass treatment in red, the random selection treatment in blue, and the control treatment in green. Upper and lower hinges of each box represent the first and third quartiles of the corresponding group. The top whisker of each box represents the highest values within 1.5 times the interquartile range and the lower whisker of each box represents the lowest values within 1.5 times the interquartile range. Each selection treatment had $n = 15$ replicate pots. Measurements for individual samples in each group are represented with points. Group means that are statistically significant from one another are labeled with horizontal bars and the corresponding $p$ value. $*p < 0.05$, $**p < 0.01$, $*** p < 0.001$.

---

**Table 1 Group means and ANOVAs for total foliar N, foliar $\delta^{15}$N vs. At. Air, % foliar N, and N agronomic efficiency for the three selection treatments at the 9th generation and group averages and ANOVAs for total foliar N, foliar $\delta^{15}$N vs. At. Air, % foliar N, and aboveground dry biomass production, for the 1st generation.**

| Generation 9 | Total foliar N (g) | Foliar $\delta^{15}$N vs. At. Air | % foliar N | N agronomic efficiency (g seed/g N added) |
|---|---|---|---|---|
| High biomass | 0.0013 $^{+/-}$ 0.00014$^a$ | 1.47 $^{+/-}$ 0.12$^a$ | 0.89 $^{+/-}$ 0.035$^a$ | 1.17 $^{+/-}$ 0.17$^a$ |
| Random | 0.0011 $^{+/-}$ 0.00013$^{ab}$ | 2.58 $^{+/-}$ 0.21$^b$ | 0.80 $^{+/-}$ 0.052$^a$ | 0.95 $^{+/-}$ 0.21$^{ab}$ |
| Control | 0.00083 $^{+/-}$ 0.000057$^b$ | 2.92 $^{+/-}$ 0.24$^b$ | 0.97 $^{+/-}$ 0.082$^a$ | 0.55 $^{+/-}$ 0.13$^b$ |
| ANOVA | | | | |
| Selection line | 0.031** | 0.0013*** | 0.22 | 0.08* |

| Generation 1 | Total foliar N (g) | Foliar $\delta^{15}$N vs. At. Air | % foliar N | Aboveground dry biomass production (g) |
|---|---|---|---|---|
| High biomass | 0.0027 $^{+/-}$ 0.65$^a$ | 5.35 $^{+/-}$ 0.65$^a$ | 2.48 $^{+/-}$ 0.50$^a$ | 0.10 $^{+/-}$ 0.019$^a$ |
| Random | 0.0027 $^{+/-}$ 0.31$^a$ | 5.39 $^{+/-}$ 0.31$^a$ | 2.04 $^{+/-}$ 0.38$^a$ | 0.13 $^{+/-}$ 0.021$^a$ |
| Control | 0.0039 $^{+/-}$ 0.56$^a$ | 5.91 $^{+/-}$ 0.56$^a$ | 2.50 $^{+/-}$ 0.35$^a$ | 0.14 $^{+/-}$ 0.023$^a$ |
| ANOVA | | | | |
| Selection line | 0.80 | 0.71 | 0.68 | 0.55 |

N data are from analysis of four replicates chosen at random from each selection treatment in the 9th and 1st generation. Average aboveground dry biomass in the 1st generation was calculated using 15 replicates from each selection treatment. The values in the table represent group means +/− SE. The superscript letters represent pairwise comparisons between selection treatments using least-squares means. *$p < 0.1$, **$p < 0.05$, ***$p < 0.01$.

---

treatments. To ensure differences observed in plant phenotype between selection treatments in the 9th generation were a result of our selection process, we analyzed aboveground dry biomass production, % foliar N, foliar $\delta^{15}$N, and total foliar N from the 1st generation (Table 1), which showed no effect of selection treatment.

To link our foliar N measurements to plant productivity, we performed Pearson correlations using data from the 9th generation of the experiment. Total foliar N was found to have a strong positive correlation with aboveground dry biomass production (df = 10, $R = 0.89$, $p = 1e-04$) (Fig. 3a) and seed yield (df = 10, $R = 0.74$, $p = 0.005$) (Fig. 3b).

**Analysis of the rhizosphere bacterial community**. We analyzed bacterial community composition through interpretations of 16 S rRNA gene sequences to assess changes in the composition and interaction profiles of microbial taxonomic groups and specific operational taxonomic units (OTU). Plotting of the Bray-Curtis distances in an ordination followed by PERMANOVA revealed

significant effects of Generation (df = 7, F-model = 17.2, $p < 0.001$), Selection Treatment (df = 2, F-model = 34.5, $p < 0.001$), and the interaction between the two factors (df = 14, F-model = 5.1, $p < 0.001$) on bacterial community composition. The PCoA shows the 1st generation samples and the entire set of control samples (1st through the 8th generation) cluster together, while the high biomass and random selection samples separate from this group together as the generations of plantings progress (Fig. 4). High biomass and random microbiomes become distinct in the 6th generation. Differences in bacterial composition between the selection treatments as the generations advanced are also displayed in the heatmap constructed using Euclidean distances and average linkage clustering (Supplementary Fig. 1).

In order to assess the effect of Generation and Selection Treatment on abundances of specific bacterial taxa, which may help explain functional shifts in the rhizosphere bacterial community, we ran ANOVAs on the relative abundances of the top seven most abundant bacterial families for the 1st to 8th generations. For our ANOVA models, we included Generation,

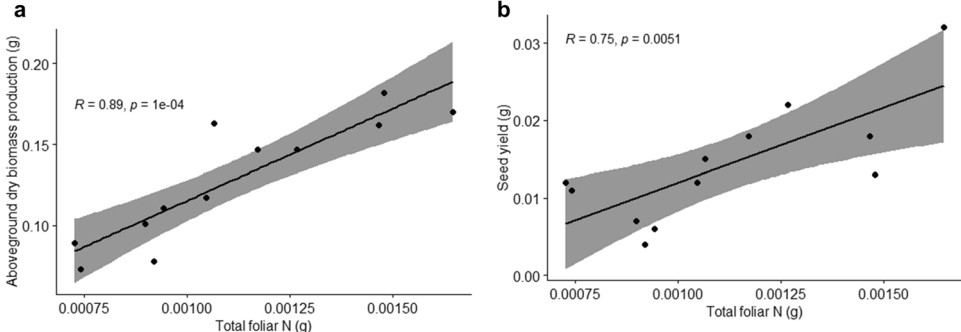

**Fig. 3 Pearson correlations between plant productivity and total foliar nitrogen (N) measurements in the 9th generation of the experiment.**
**a** Scatterplot and Pearson correlation of aboveground dry biomass production (g) vs. total foliar N (g). **b** Scatterplot and Pearson correlation of seed yield (g) vs. total foliar N (g). Each point represents the seed yield/productivity data and total foliar N data of a sample from the 9th generation. The line represents a linear regression between the two variables in each graph. The shaded region represents the 95% confidence interval in each graph. Each selection treatment had $n = 4$ replicates for this analysis.

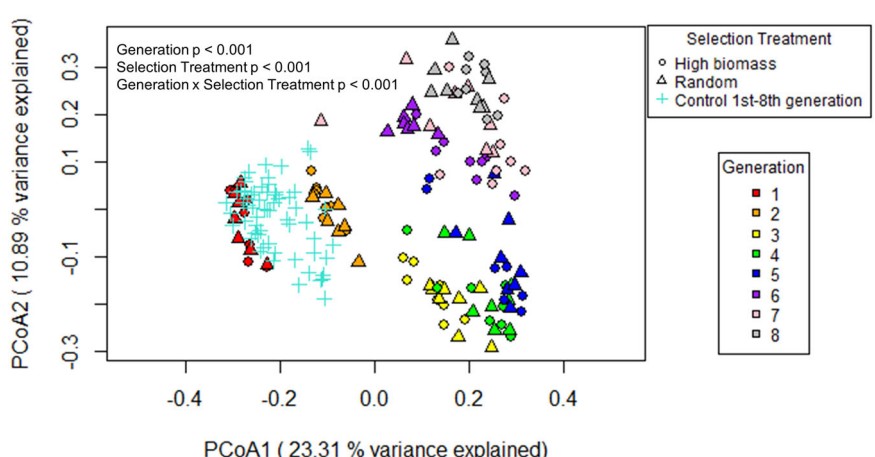

**Fig. 4 Principal coordinates analysis (PCoA) of the rhizosphere microbial community based on 16 S rRNA gene sequencing.** Circles indicate microbiomes from the high biomass selection treatment and triangles indicate those derived from random selection. The crosses represent the control selection treatment, which are used to measure against genetic drift and growth chamber adaptation. For the high biomass and random selection treatments, the cycles of plantings are represented by color: 1st through 8th are red, orange, yellow, green, blue, purple, pink, and gray, respectively. For the control selection treatment, all generations (1st through 8th) are represented in teal. Generation x Selection Treatment groups had at least seven replicates. In total, sequences from 186 soil samples were used in this analysis. PERMANOVA revealed significant effects of Generation ($p < 0.001$), Selection Treatment ($p < 0.001$), and the interaction between the two factors ($p < 0.001$) on bacterial community composition.

Selection Treatment, and the interaction between the two as model factors, then report here pairwise comparisons of the abundances of different bacterial families in the 8th generation between the selection treatment. In other words, the ANOVAs and pairwise comparisons reported here are from analysis of the Generation x Selection Treatment interaction term. Our analysis shows shifts in the quantities of bacterial families such as *Chitinophagaceae* (df = 14, $F$ value = 1.97, $p$ value = 0.022) (Supplementary Fig. 2). Members of this family, which contains plant growth promoting species such as *Arachidicoccus rhizosphaerae*[29], increased in relative abundance in the high biomass and random selection treatments by 378.1% ($p = 0.0007$) and 329.5% ($p = 0.026$) compared to the control treatment in the 8th generation (Supplementary Fig. 2). Our analysis of *Flavobacteriaceae* (df = 14, $F$ value = 4.74, $p$ value = 2.98e-7), which is a member of the widespread *Bacteroidetes* phylum, and *Oxalobacteraceae* (df = 14, $F$ value = 5.46, $p$ value = 1.6e-8), belonging to the functionally diverse *Proteobacteria* phylum, revealed further shifts in abundances between selection

treatments in the 8th generation. The abundance of *Flavobacteriaceae* was 83.1% lower in the high biomass selection treatment ($p < 0.001$) and 84.2% lower in the random selection treatment ($p < 0.001$) compared to the control treatment. The abundance of *Oxalobacteraceae* was 51.1% lower in the high biomass selection treatment ($p = 0.0082$) and 60.6% lower in the random selection treatment ($p = 0.0006$), compared to the control treatment. Analysis of *Sphingobacteriaceae*, also a member of the *Bacteroidetes* phylum, revealed further shifts between selection treatments in the 8th generation (df = 14, $F$ value = 5.61, $p$ value = 8.69e-9). When analyzing *Sphinobacteriacaea* abundance between selection treatments in the 8th generation, we found the high biomass selection treatment had 52.3% lower abundance compared to the random selection treatment ($p < 0.001$) and 37.6% lower abundance compared to the control treatment ($p = 0.0136$).

Analysis of Shannon diversity indexes revealed differences in bacterial diversity between all three selection treatments (df = 2, $F$ value=14.7, $p = 1.12e-06$) (Fig. 5). The control treatment had

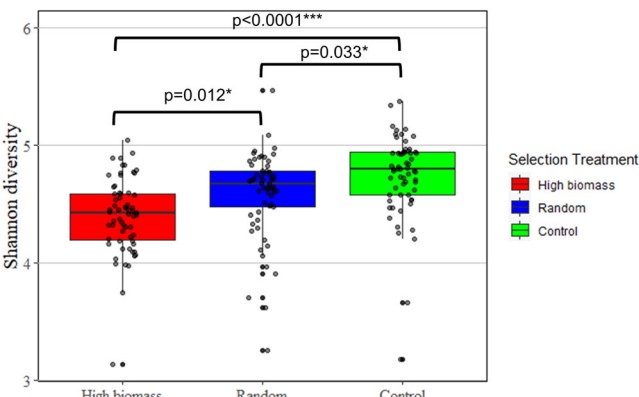

**Fig. 5 Analysis of mean Shannon diversity index for the three selection treatments using microbial data from all eight generations of soil samples sequenced.** The different selection treatments are represented in color with the high biomass treatment in red, the random selection treatment in green, and the control treatment in blue. Upper and lower hinges of each box represent the first and third quartiles of the corresponding group. The top whisker of each box represents the highest values within 1.5 times the interquartile range and the lower whisker of each box represents the lowest values within 1.5 times the interquartile range. Measurements for individual samples in each group are represented with points. Most Generation x Selection Treatment groups had eight replicates each after quality control. In total, sequences from 186 soil samples were used in this analysis. Group means that are statistically significant from one another are labeled with horizontal bars and the corresponding *p* value. *p < 0.05, **p < 0.01, ***p < 0.001.

the greatest bacterial diversity, with a diversity index 3.4% greater than the random selection treatment ($p = 0.033$) and 7.7% greater than the high biomass selection treatment ($p < 0.0001$). In addition, the random selection treatment had a diversity index 4% greater than the high biomass selection treatment ($p = 0.012$). This decrease in diversity could be indicative of ecological filtering[30] or a similar process in the high biomass selection treatment.

**Microbiome network analysis.** To compare how bacterial group dynamics shifted between the different selection treatments in our experiment, we performed an Extended Local Similarity Analysis (eLSA) using bacterial sequence data from the 1st through 8th generation. Briefly, an eLSA is a modified version of Local Similarity Analysis (LSA), which measures local and potentially time-delayed co-occurrence patterns in time series data[31]. Unlike LSA, eLSA can reveal statistically significant local and potentially time-delayed association patterns in replicated time series data, such as the bacterial sequence data from this experiment. Results from the analysis can then be used to construct an association network consisting of nodes (representing bacterial Operational Taxonomic Units, or OTUs), edges (representing potential relationships between nodes), and modules (clusters of nodes), which together can be used to visualize changes in microbial interaction networks (i.e. microbial ecological relationships).

The eLSA revealed that the three selection treatments differ in edge connections with the same node set, which included OTUs that had a relative abundance above 1% in at least 1 sample. The high biomass selection treatment showed the greatest number of edge connections, followed by the random selection treatment, then the control treatment (Fig. 6a). Notably, the high biomass selection treatment visually showed a densely connected cluster of OTUs not observed in the random or control selection treatments when the three networks are stacked (Fig. 6b). When plotting the

degree distribution (number of edges connected to a specific node) of all three selection treatments, the high biomass selection treatment had substantially more nodes that are more connected when the degree is larger than 12 (Supplementary Fig. 3a). Changing the alpha threshold for the eLSA analysis, which determines if a correlation between two OTUs is significant, yields consistent results where the network of the high biomass selection treatment is always the most dense (Supplemental Fig. 3b) and indicates robustness of eLSA on this data set.

In addition to differences in edge connections between the three selection treatments, we found decreased modularity of the bacterial community under high biomass selection, which is indicative of a more highly organized network topology displaying tightly connected bacterial OTUs. The high biomass bacterial communities showed the lowest modularity of 0.546, followed by 0.562 in the random selection group and 0.741 in the control group. Module size, taxonomic composition, and the interaction strength (connectivity, as indicated by the node size) differed across the selection treatments and control. Figure 6c shows the two largest group formations within the network and their taxonomic profile. The greater density of the two major groups in the high biomass selection treatment compared to those generated from random selection shows stronger interconnection within the groups.

To further quantify these visual differences, we used our sample edge lists to run a publicly available network model tool, known as NetShift[32], which quantifies bacterial community rewiring and changes between two data sets. With the analysis, we sought to perform pairwise comparisons of global graph properties for the three selection treatments. Our analysis revealed the high biomass selection treatment overall had higher network density (portion of potential connections in a network that are actual connections), lower average pathlength (lower average number of steps connecting one node to the next), more total edges (more connections), and more exclusive edges than the random selection and control treatments (Fig. 6d). The results indicate a more connected bacterial community in the high biomass selection treatment compared to the random and control treatments, which was also visually observed in our microbial network constructed from the eLSA.

## Discussion

The objective of the experiment was to examine changes in rhizosphere bacterial group dynamics across generations of selection for high biomass growth of a non-mycorrhizal plant under soil nitrogen limiting conditions. In essence, we used a specific phenotype of the host plant (high aboveground biomass) to report the collective actions of a bacterial community that are modifiable over the course of selection[4]. We used an ecological filtering approach[30] in this experiment to remove rhizosphere microbiota that were not associated with the four highest performing units for each generation of plantings in the high biomass selection treatment. The plants used in this experiment did not undergo selection pressure. Instead, seeds of the same parental origin were used across generations of plantings, which means the genetic profile of plants from the 1st through 9th generations were the same. The PCoA indicated divergence in the bacterial communities between the high biomass and random treatments starting in the 6th generation. The bacterial network analysis using eLSA and NetShift indicate that bacteria associated with increased aboveground biomass may be coordinating group-level behaviors that impact fitness traits, such as the higher seed yield in the high biomass selection treatment compared to the control treatment. Altogether, the stronger connectivity of specific taxonomic groups forming within the rhizosphere of the high biomass plants over

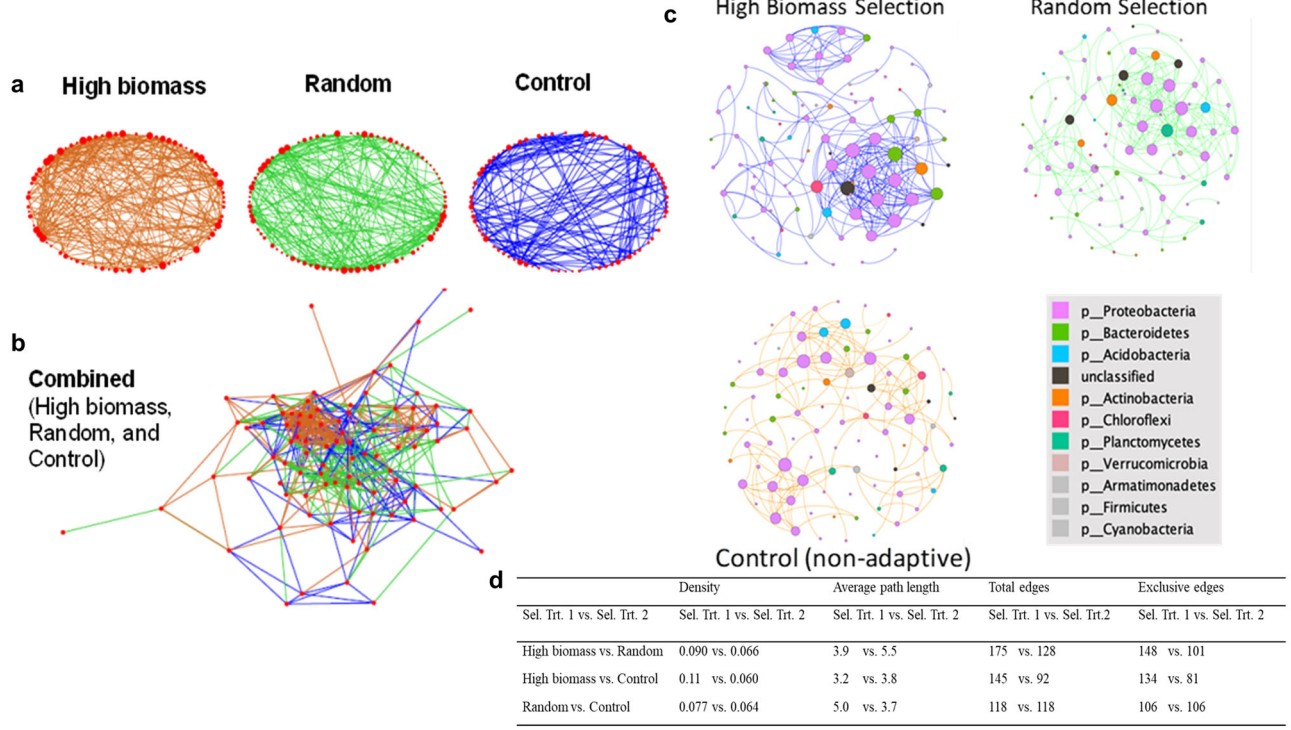

**Fig. 6 Extended Local Similarity Analysis (eLSA) networks representing microbial taxa (OTU) interactions. a** Networks of the three different selection treatments comprised of sequencing data from all eight generations. Size of the node is proportional to the degree of the node (the number of connections a node has). **b** Network of the combined three selection treatments. **c** Two largest modules (clusters) from each selection treatment and control using our network data. The networks represent OTU interactions for different selection treatments using sequencing data from all eight generations. Size of the node is proportional to the level of connectivity and colors represent taxonomic groups of bacteria. **d** Comparisons of network density, average path length, total edges, and exclusive edges between selection treatments using NetShift. We used the same edge lists generated for our LSA to run our NetShift analysis. The high biomass selection treatment had 63 replicates, the random selection treatment had 61 replicates, and the control treatment had 62 replicates. In total 186 soil samples were used for this analysis.

time, along with the observed changes in bacterial community composition, may have contributed to shifts in plant seed yield and plant N dynamics.

The eLSA network model revealed that the selection process had a significant effect on bacterial group dynamics in each treatment over the course of the experiment. A notable outcome of the experiment was the formation of bacterial groups (modules) that differed in composition and interaction intensity across the selection treatments. Modules or clusters are biologically relevant units whose interaction patterns provide information in understanding the function of the network[33]. For example, past research has suggested highly modular microbiota could play a key role in plant immune responses to pathogens[34]. The significance of these groups is how they may reveal emergent properties of microbiomes associated with alterations in the host's phenotype, such as seed yield. In this study, the selective pressure on plant biomass resulted in lower diversity and higher connectivity in the high biomass selection treatment, suggesting a highly organized bacterial community formed in the rhizosphere.

Analysis using NetShift provided quantitative data confirming changes in bacterial community dynamics across treatments. NetShift indicated a more interconnected bacterial community: lower average path length, which describes the average number of steps connecting one node to the next, and total edges, which represents connectedness among community members[32]. Previous work has highlighted the influence of external pressures on bacterial community properties, such as density. Faust et al.[35] showed that microbial networks in bulk soils are generally less dense than host-associated networks. Considering empirical

results such as this, it is conceivable the selection pressure for increased plant biomass production resulted in highly connected assemblies of bacterial groups that coordinate complex functions that are beneficial to plant growth and N use efficiency.

It is well known that the rhizosphere bacterial community regulates several plant activities, such as nutrient uptake, through a variety of mechanisms. For example, rhizobia form nodules in legume roots and fix atmospheric nitrogen for their host[36], while free-living microbes, such as phosphorus solubilizing bacteria, regulate plant P availability through collective production of extracellular enzymes[37]. It is possible that the changes observed in plant tissue N content in this experiment were a result of changes in the nutrient cycling activities of the rhizosphere bacterial community between the selection and control treatments. Our sequencing data indicate that the bacterial communities of the high biomass selection treatment showed distinct characteristics from the bacterial communities of the random selection and control treatments. Among these distinct characteristics were a decrease in bacterial diversity, which could indicate ecological filtering[30] for more efficient N cycling microorganisms. An example is the shift in the relative abundance of *Chitinophagaceae* bacteria, which have previously been shown to be associated with increased N cycling activity[38].

By repeatedly selecting for rhizosphere bacterial communities associated with greater aboveground biomass production, it is conceivable the rhizosphere bacterial communities in the high biomass selection treatment became enriched with taxa that aid in the mineralization of N, which is an essential nutrient for seed production and development[39–41]. It is important to note

that the plants in this study were grown under nitrogen limitation, specifically 20 ml of 100 ppm N fertilizer as one dose in each generation. In nitrogen-limiting conditions such as this, plants may enrich the rhizosphere for N capturing microorganisms via root exudation[42], which could impact soil N availability or N use efficiency in the host.

Stable isotopes provide an opportunity to assess potential contributions of the rhizosphere microbiome to plant N utilization[43,44]. Changes in foliar $\delta^{15}N$ result from corresponding alterations in the $\delta^{15}N$ of the soil solution or some fundamental change in within-plant N metabolism, which is often interpreted as modifications in soil processes, especially nitrification and denitrification[45]. In this study, it is challenging to determine the soil or metabolic processes that were impacted between the 1st and 9th generations. However, the observed changes in foliar $\delta^{15}N$ (Table 1) suggest some alteration in soil N cycling by the rhizosphere microbial community or metabolism in the high biomass selection treatment. There are several potential ways in which the rhizosphere could change soil N cycling. For example, differences in foliar $\delta^{15}N$ values among the treatments could be a result of differential rates of nitrification and denitrification between groups[46]. Similarly, differences in $\delta^{15}N$ values could indicate contrasts in fertilizer- versus soil-N use by plants across treatments[47], which could be mediated by the microbiome. Plant fractionation of nitrogen isotopes may have differed across selection treatments[48]. The increases in total foliar N and N agronomic use efficiency, and the $\delta^{15}N$ data altogether suggest alterations in rhizosphere N cycling and plant N utilization in the high biomass selection treatment. As shown in our regression analysis, this alteration in available N could be a major mechanism by which bacterial communities in the high biomass selection treatment altered the yield and NUE phenotypes of *B. rapa*.

In addition to potentially promoting N cycling, the rhizosphere bacterial community in the high biomass selection treatment could have also promoted the uptake of other important plant nutrients, such as phosphorus[49,50], but plant tissue P and soil extractable orthophosphate were not analyzed in this study. It is also possible the shifts in the high biomass bacterial communities over the course of selection represent the elimination of organisms that inhibit plant productivity[51], as we observed a decline in diversity in the high biomass selection treatment compared to the other two treatments. Our analysis revealed lower abundances of taxa belonging to the family *Sphingobacteriaceae* in the high biomass treatment. This lower abundance may have benefited plants in the high biomass treatment as members of this family have been associated with plant and animal diseases[52,53]. We expected ecological filtering of the bacterial community, and thus a decline in diversity as observed in our experiment, since our selection for high biomass production likely altered bacterial community assembly processes[54,55].

We have shown in this experiment that it is possible to develop rhizosphere bacterial communities associated with desirable plant traits in agriculturally relevant crops, such as *B. rapa*, through artificial group selection over multiple generations. Our results show seed yield and plant tissue N may have been altered by the microbiomes assembled under high biomass selection. While both high biomass and random selection treatment plants had the same aboveground biomass and the soil microbial communities were largely similar, the main conclusion from this study indicates that iterative selection over time shows divergences in microbiome composition that may be associated with increased seed yield and nitrogen use efficiency. Our work demonstrates that group-level behaviors within the rhizosphere can be altered under selection and may serve a key role in modifying plant host phenotypes related to productivity. This project contributes to the growing area of research showing that rhizosphere microbiomes influence crop traits and have the potential to promote plant productivity and yield[56,57] without the use of genetic engineering or gene-editing techniques. As this experiment focused on bacterial communities, future studies could be designed to analyze how microbiome composition or behaviors can be modified to alter plant fitness and other phenotypes. These types of microbial experimental systems could ultimately help develop a better understanding of plant-microbe interactions that can be utilized for regenerative agriculture.

## Methods

**Growth chamber conditions**. All plants in this experiment were grown in a growth chamber set at 350 μmol of light, 10% humidity, alternating between 28 °C for 16 h and 23 °C for 8 h for every generation (Percival-Cornell University Weill Hall Life Sciences Growth Chamber Facility, Ithaca, NY, USA).

**Selection on rhizosphere microbiomes**. Bacterial inoculants for the high biomass and random selection treatment were created using an iterative selection process over nine generations adapted from Swenson et al.[10] and Panke-Buisse et al.[11]. In the 1st generation, 0.5 L pots were filled with LM-111 container soil (Lambert Peat Moss, Inc., Riviere-Ouelle, Quebec, Canada) that was autoclaved twice with a 24-hour resting period in between cycles. After sterilization, the pots were inoculated with 20 mL of a bacterial inoculant created from field soils gathered from diverse organic farms in Ithaca, New York (42.4440° N, 76.5019° W). This initial bacterial inoculant was created by compiling and homogenizing the field soils, then adding 55 mL of the mix to 500 mL of sterilized deionized water. The mix was then shaken with 30 mL of sterilized glass beads at 180 opm for one hour. The resulting slurry was then filtered through four layers of sterilized food preparation cloth, which retains a large portion of the soil bacterial community in the filtered slurry while eliminating large soil particles. Aliquots of this mix were frozen at −20 °C to serve as bacterial inoculants for the control selection treatment in the following generations. In every generation, all three selection treatments (high biomass, random, control) had 15 replicate pots total. The high biomass and random selection inoculants were kept frozen at −20 °C for two days and thawed, along with the control inoculants, prior to the start of each generation. This additional process of freezing and thawing at each generation was set in place to minimize potential confounding differences between the control samples and the selection treatments.

A single *B. rapa* seed pool consisting of a single genotype was gathered for use throughout the entire experiment. This prevented any genetic changes at the plant host level, helping to ensure any changes in plant phenotype observed in our experiment were a result of changes in the bacterial community. Before planting, seeds were surface sterilized with 2.5% bleach for 15 min and then rinsed with sterile water at least five times before seeding. Eight seeds were planted into each pot to ensure adequate germination. Pots were then thinned to five plants per pot upon emergence. Pots were placed randomly into trays covered with sterile clear plastic domes to help prevent outside contamination. The trays were arranged randomly across two growth chamber benches. Plants were watered daily with filter-sterilized water and received one dose of 20 mL filter sterilized Jacks 21-5-20 + Epsom salt fertilizer, at a concentration of 100 ppm N. Plants were grown for 10 days, then the stem and leaf aboveground biomass of all plants in each pot were harvested by cutting at the base of each plant's cotyledon leaves. The composited plant samples from each pot were dried at 65 °C for 24 h to record total aboveground dry biomass production (i.e. the biomass reported for each pot is the biomass production of all plants, not an average).

Rhizosphere soils were collected and stored at 4 °C upon harvest. The soil was collected by gently breaking up the root ball, holding the roots, and gently shaking away unattached soil. For the high biomass selection treatment, we kept soils from the four pots with the greatest total aboveground biomass production (summed total of the plants within the pot) to create the high biomass treatment microbiome inoculant for the following generation. After creating a homogenous mix of the four soils, 100 mL of the soil mix was dissolved into 500 mL of sterile DI water. The slurry was then shaken with 30 mL of sterile glass beads for one hour at 180 opm and filtered through four layers of sterile cheesecloth. The resulting mix was then used as the bacterial inoculant for plants in the successive generation of the high biomass selection treatment with each pot receiving 20 mL of the inoculant. For the random selection treatment, the rhizosphere soil from four random plants (selected using a random number generator on Microsoft Excel) was used to create bacterial inoculants for the next generation of plants in this selection treatment. The four randomly chosen soils were homogenized and 100 mL was then mixed with 500 mL of sterile DI water, shaken at 180 opm with 30 mL of sterile glass beads for one hour, then filtered through four layers of sterile cheese cloth. This filtrate was then used to inoculate plants in the random selection treatment in the following generation with each plant receiving 20 mL of the mix. Soils from eight random plants were kept from the control selection treatment, which were only used for analysis of the rhizosphere bacterial

community (i.e. they were not used to make bacterial inoculants for the following generations). We repeated this exact selection process for a total of 9 generations (Fig. 1).

At the final generation (9th generation), we grew the plants to seed in order to assess the effect of our selection process on *B. rapa* seed yield, which required plants to be grown for 31 days. Upon flowering (~18 days), plants were hand pollinated daily by transferring pollen to neighbor *B. rapa* plants. In order to initiate seed ripening, we ceased watering the plants two days before harvesting. After harvesting, stem and leaf tissue was collected and dried at 65 °C for 24 h then weighed to record total dry aboveground biomass production. Seeds were collected from all plants in every pot and weighed to assess seed yield. Because the plants required dried soil to set seed in this generation, we were not able to collect soil samples for sequencing the rhizosphere bacterial community for this final time point.

**Bacterial 16 S rRNA gene sequencing**. Bacterial DNA was extracted from frozen rhizosphere soil samples collected from the 1st through 8th generation using a PowerSoil DNA Isolation Kit (Qiagen, Hilden, Germany) following the manufacturers protocol for more absorbent soils. Approximately 150 to 200 mg of soil from each sample was used for extraction. 16 S rRNA gene sequences were amplified using PCR primers 341 f (50-195 CCTACGGGNGGCWGCAG-30) and 805 R (50-GACTACHVGGGTATCTAATCC-30)[58], which target the bacterial/archeal 16 S rRNA gene variable region for pair-end Illumina (Illumina, Inc., San Diego, CA, USA) barcoded sequencing. Each selection treatment in the 1st generation had one sample with poor PCR amplification, which were excluded from downstream analyses. After an initial clean-up of the successfully amplified samples using the HighPrep PCR Clean-up System (MAGBIO Genomics, Gaithersburg, MD, USA), unique Index Primers were attached to amplicons in each sample with a second PCR cycle. The indexed samples were cleaned and normalized using the SequalPrep Normalization Plate Kit (Thermo Fisher Scientific, Waltham, MA, USA). Sample normalization was followed by pooling 5 ul of each sample into one composite sample. The pooled sample was then run on a 1.2% agarose gel with SyberSafe added and the target band was excised. The DNA was then extracted from the gel using the Wizard SV Gel and PCR Clean-Up System (Promega, Madison, WI, USA). The final pooled sample was then sent to the Cornell Genomics Facility (Cornell University, Ithaca, NY, USA) for sequencing on the Illumina MiSeq platform using the v3 paired-end 300 bp kit.

**Plant tissue N analysis**. N content and isotope ratio of plant tissue in this experiment was analyzed at the Cornell Stable Isotope Laboratory (COIL) using a Thermo Delta V isotope ratio mass spectrometer (IRMS) interfaced to an NC2500 elemental analyzer. Tissue from four random samples from each Generation x Selection Treatment group were submitted to COIL for analysis. Measurements of %N were used along with aboveground dry biomass production to calculate total plant N.

**Statistics and reproducibility**. All statistical analyses were performed using R statistical software (Rproject.org). To analyze the effect of our selection process, we analyzed plant biomass production, seed yield, and N data from the 9th generation using linear modeling. We focused on plant data from this generation and analyzed it separate from other generations given it was the final generation of the experiment and was grown for a longer period than the generations before. One-way ANOVAs were performed for each model to determine the effect of selection treatment on each outcome in the 9th generation. Least squared means was then used to perform pairwise comparisons of group means between selection treatments. Normality of residuals and homoscedasticity were verified for each model using plotting methods. To show our selection process was responsible for any differences observed in our ANOVAs at the 9th generation, we performed these same procedures on data from the 1st generation, which showed no differences in treatment means for any response variable (Table 1). Data transformations were performed as necessary for all analyses to meet ANOVA assumptions.

Pearson correlations were performed using foliar N data, aboveground dry biomass production, and seed yield data from the 9th generation to assess potential linkages between plant nitrogen uptake and productivity. Data were first visualized in scatterplots using the ggscatter function to verify linear relationships between variables. Normality for each variable was then verified using the Shapiro-Wilk normality test. We then ran our correlation analysis using the cor.test function in R, which provided Pearson correlation coefficients along with *p* values to indicate if a relationship was significant.

**Analysis of microbiome sequences**. The pipeline from the Brazilian Microbiome Project (http://www.brmicrobiome.org/) was modified to process our sequences. Mothur v. 1.3613 was used to merge paired-end sequence (make.contigs), trim off primers (trim.seqs, pdiffs = 2, maxambig = 0), remove singletons (unique.seqs → split.abund, cutoff = 1) and classify sequences (97% similarity). OTUs that were suspected to not be of bacterial origin were removed (remove.lineage). Qiime v.

1.9.114 was used to cluster OTUs and create an OTU table. Raw sequence data are available via the National Center for Biotechnology Information (https://www.ncbi.nlm.nih.gov/) repository under accession PRJNA833111[59].

Following sequence processing, the dataset was rarefed to 2988 sequences per sample (i.e. the minimum number of reads per sample in this dataset). We then converted OTU counts in each sample to percentage, then calculated Bray-Curtis distances using vegdist. We then performed a PCoA analysis on the Bray-Curtis distances calculated using cmdscale. We then calculated % variance explained by the first two axes, then plotted the calculated Bray-Curtis distances in an ordination to visualize the effect of Generation and Selection Treatment on community composition. The adonis package was used to perform a PERMANOVA on our sequence data and assess the effects of Generation, Selection Treatment, and the interaction between the two factors on community composition. Due to a lack of adequate post hoc tests in adonis, we were not able to do pairwise comparisons. PERMANOVA requires homogeneity of variance, which we analyzed using the betadisper function. Shannon diversity indices were calculated for each sample using the diversity function. We then created a linear model assessing the effect of Selection Treatment alone on the Shannon diversity indices of our samples (i.e. Generation and an interaction term were not included in the model), then ran an ANOVA on the model. We then used least squares means to do pairwise comparisons of diversity indices.

Changes in the relative abundances (%) of different bacterial taxa across generations and treatments were presented with stacked barplots (Supplementary Fig. 2). We aggregated the OTU, taxonomic, and metadata using the phyloseq function, then identified the seven most abundant bacterial families using the sort function. We then calculated average % abundance for each bacterial family across our Generation x Selection Treatment groups then plotted the abundances in a stacked barplot. ANOVAs were then performed on the relative abundances of each taxonomic group using linear modeling and group means were compared using least squared means to quantitatively assess differences in taxonomic abundances across generations and selection treatments. Normality of residuals and homoscedasticity were verified for each model using plotting methods, and transformations were performed as necessary to meet ANOVA assumptions. A heatmap was also created to visualize changes in the relative abundances of bacterial taxa. We first identified the maximum abundance of each individual OTU in every sample, then filtered out OTUs with maximum abundances below 7.5%, which we found was an appropriate cutoff to visually see shifts in OTU abundances across generations and selection treatments in our heatmaps. After filtering, we then used the pheatmap function in R to visualize shifts in OTU relative abundances across generations and selection treatment using Euclidean distances of OTU relative abundances and average linkage clustering.

**Extended local similarity analysis (eLSA) and network construction**. In order to study the effect of our selection process on bacterial community co-occurrence patterns, which can represent important ecological interactions and processes[60], we constructed dynamic networks for each selection treatment using bacterial sequence data from the 1st through 8th generation. To construct our networks, we employed Extended Local Similarity Analysis (eLSA), a similarity-based method that uses dynamic programming to build association networks from time series data[31,61]. The algorithm used is summarized in Supplementary Table 1. In a standard LSA method, we are provided sequence data in which only one sample is available for each time step for each sequence we have. We first used pairs of OTU time series data of the same length as inputs. Positive and negative correlation scores between each pair were then calculated, which we used to determine local similarity scores. Larger local similarity scores indicated stronger potential relationships between the two paired OTUs. However, this is too stringent in our case where multiple samples are available within one sequence for each time step. Hence, instead of using LSA, we applied the eLSA method in this work where a sufficient statistic is computed for each time step given all the samples observed at that time step and then applied the standard LSA method to compare the local similarity score. Here, we used the simple average method to summarize the replicated time series data[31]. For example, if we have 10 samples at each time step, we can then compute the average value of the 10 samples at the same time step and apply LSA on this new sequence data where the data at each time step is the average value for all samples. In this paper, we take the mean of all the samples observed at every time step.

Next, we used our LSA scores between OTUs to construct a network representing the strong relationships within bacterial communities. We defined a graph G (V, Ɛ) where V = {v1, v2,…, vm} represents different OTUs and (vi, vj) ∈ Ɛ if there is a strong correlation between OTUs represented by vi, vj. Strong correlations here are defined using the significance level α.

To compute *p* values for two OTU time series data O1 and O2, we first randomly permuted O1 and O2 and computed LS (O1, O2). *P* value here was defined as the probability that this local similarity value is at least as large as the local similarity value with the non-permuted data. We computed an upper bound for the *p* value[62]. After establishing the upper bound for the *p* value between O1 and O2, we defined a global significance level α and filtered the upper bounds with α. Specifically, if the *p* value upper bound between OTU time series represented by vi and vj was larger than α, we drew an edge between vi and vj. For our analysis,

only OTUs with a relative abundance of at least 1% in any of the samples were used, which gave us a network with 90 nodes representing different OTUs. We used a time delay of 1 and the significance level α to be 1. The code used for this analysis is available via https://doi.org/10.5281/zenodo.6800595[63].

**Reporting summary**. Further information on research design is available in the Nature Research Reporting Summary linked to this article.

## Data availability

The raw bacterial sequencing dataset are available in the National Center for Biotechnology Information Sequence Read Archive, https://www.ncbi.nlm.nih.gov/sra/PRJNA833111. The corresponding accession code is PRJNA833111. Data used to generate Figs. 2 and 5 are included in Supplementary Data 1.

## Code availability

For the network model, we used a custom code to run the Extended Local Similarity Analysis (eLSA) discussed in this paper. The code can be accessed via https://doi.org/10.5281/zenodo.6800595.

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

## Acknowledgements

The authors would like to thank Andrew Pochedly, Micaela Moravek, Kim Sparks, Lynn Johnson, Olivia McCandless, Emaleigh Perry, Ololade Olawale, and Michelle Chen for help with the research. This work was supported by an Agriculture and Food Research Initiative Grant [2016-67013-24414] from the United States Department of Agriculture National Institute of Food and Agriculture. J.G. was supported by fellowships from the National Science Foundation Graduate Research Fellowship Program (NSF-GRFP) [DGE-1650441] and the Cornell University McNair SUNY Diversity Fellowship.

## Author contributions

J.K. and M.G. designed the experiment. M.G. and L.W. conducted the experiment. J. Garcia, L.C., S.H., and J.S. analyzed the data. J. Garcia and J.K. wrote the manuscript. All authors, including J. Giovannoni, contributed to final manuscript preparation and revisions.

## Competing interests

The authors declare no competing interests.
