## [Peer Review File · Communications Biology]

Reviewers' comments:

Reviewer #1 (Remarks to the Author):

Title: Emergent Network of a Modifiable Plant Phenotype Reveals High Connectivity of Microbiome Members

Summary: The authors describe a study where they applied selective pressure on the soil microbiome for changes in biomass phenotype through 9 generations of *Brassica rapa*. They then characterize the microbial community from generations 1 through 8, showing a decrease in diversity due to microbiome selection but increases in connectivity in the high biomass selection line.

Overall comments:

While I support the hypotheses being tested could be answered through the experimental and methodological approach, I feel like conclusions drawn about differences in microbial function and their influence on plant phenotypes are not strongly supported. However, there is microbial community selection across generations as high biomass effects on plant phenotypes differ from control. Specifically, what is the importance of the random selection line? Improved description of these treatments is imperative.

I also found the co-occurrence/network analysis on the soil microbiome to be appropriate and informative. While the approach is unique and provides interesting detail, authors should be careful as they try to make conclusions on microbial function based on diversity and co-occurrence patterns alone.

While most of the manuscript was well written I believe there is room for improvement, particularly by holding the reader's hand in the shorter "results" first format and in being more specific and less speculative in the discussion. For example, I believe there needs to be some clarifying methods text throughout the results section that provides enough necessary context for the reader to understand the results as they are presented. Similarly, there are several areas where main conclusion paragraphs feel terse, out of place, or largely speculative. This makes the discussion hard to follow and detracts from the main results and conclusions.

Comments by line:

Line 74-76: No hypotheses or expectations for potential patterns in biomass and seed yield? Confusing as written....is there a hypothesis or not?

Line 80: I don't think is the place to outline the importance of these 3 treatments, but this needs to come out early in the results section so the reader can understand the results in the context and reason for each of the specific selection line and control treatments. Especially the random and control treatment? What do these treatments provide? The entire study hinges on these treatments and the reader needs to understand this upfront.

Line 88 and throughout Results: I feel it is necessary to move some brief methodology into each Results section. The "results" first style requires some well-placed text to provide context for the results being presented. Especially some of the experimental design, descriptions of selection lines, etc.

Line 92: Is the control treatment just the original field-collected and prepped inocula? Might want to clarify these types of delineations to help the reader with the main results.

Lines 94-100: Here is the initial result showing high biomass phenotypes don't differ from those in random selection lines. Likewise, Table 1 shows the lack of differences in the measured plant phenotypes, aside from the isotope ratio.

Lines 103-104: It is hard to determine here if "selection line" and "treatment" are the same thing or something different.....needs to be clarified or rewritten to improve understanding.

Lines 120-122: Is the main takeaway here that no differences were determined after the first-generation harvest? Which should all be the same inocula, and you shouldn't expect differences after generation 1? Or that through 9 generations of the selection lines and control treatments we finally start to see divergence of the selection line treatments from the control treatment in terms of community-level selection of the microbiome? Clarifying the text about expectations in the introduction and the results could be very helpful.

Lines 136-157: Are the described patterns determined from comparing generations 1 and 8? Or across all generations? How was this done? Also, there should be some reference to the supplemental figure of the stacked bar plots of relative abundances?

Line 159: Since there is microbial data for each generation, why not determine community turnover through time?

Lines 166-211: Very cool work here.

Lines 257-264: This paragraph feels out of place. The last two sentences specifically don't relate to findings from the study and seem misleading. Consider reworking this paragraph to make the intent clearer.

Lines 272-277: These lines are confusing and don't make a concise point? What about your ordination and analysis are you alluding to? What does a decrease in bacterial diversity really tell us?

Lines 304-312: These sentences are incredibly speculative and do more harm than good. Overall, a large portion of the discussion feels speculative and not reinforced with specific results from the study.

Line 445: "fungal"? Don't think fungi were characterized or analyzed in this study. But since it was left in there....why weren't fungi included in the study? They obviously have a direct influence on plant productivity and resource use.

Lines 458 – 465? I know these plots are in the supplement, but where are they referenced in the main manuscript?

Reviewer #2 (Remarks to the Author):

This is a captivating manuscript on selection of soil bacterial communities based on plant performance and hence on creating/steering microbial communities. The experiment is clever and well conducted and thought through and manuscript well written. All together very clever!

There are, however, few aspects that would make this article more appealing and help it to attract all the attention the work deserves. First of all, the title is not very informative on the content. Actually, I would not probably read article with this title. To me, more important findings are the selection of microbiomes for plant performance (and maybe a bit the network aspect). This could be better reflected in the title of the paper. (The sentence in lines 18-19 would reflect better what was done). Furthermore, the figures let one down a bit. It would be useful to have a clear figure on how

experiments were performed and this could shorten a bit the text on it. In Fig. 1 I would prefer to see boxplots (for variation and sample number) and colors would be also more pleasant. In Figure 2 the colors should be in legend and markers bigger. Colors could be also a scheme from light to red to see clearly the generation differences. Fig. 4 (the key finding): it is not very clear from this figure that there are more connections in high biomass systems (main message). I would say that random looks very similar to the high biomass systems. Please add stats/clearer representation of the data. Supplementary fig 4 is much clearer but indeed does not show all data.

The experimental set-up is in large part well explained. Some things are a bit unclear still. There were 5 plants per pot and 15 replicates. You used the average number of the 5 plants? And only above-ground biomass was measured? How was the random sample determined? i.e. the random sample based on randomness could be theoretically the same samples as high-performers. This would be interesting to see and discuss (also in terms of calculated variance).

One would expect the variation to become larger in random selection across soils but this is not looked at (in microbiomes or plants) but might be interesting. It would have been also interesting to look at if selecting for the other extreme (small plants) would lead to similar network pattern compared to random. This cannot be changed of course but can be briefly discussed.

Also, only bacteria that were even filtered were used and studied in the experiment, how would conclusions change if more realistic systems including other microbes and even larger organisms would be included? Microbiome is not only bacteria and hence it is preferred to use exact terms also here.

The statistical models are quite simple (ANOVA) and one should note that most bacterial taxa are not distributed in the way that would allow these analyses. Please consider checking the assumptions (and variances) before presenting information on these groups and use correct models if assumptions are violated.

The heatmap creation should be specified and it could be noted that they are not overly informative here.

Why was filtering for OTUs set to 1% in any sample? This will include only very dominant species when diversity is high.

It is also shame that the experiment was not longer and that seed biomass could not be measured earlier as now it is not clear when it was changed.

Abstract line 19-20 this sounds like the 9th generation is the key and this would only happen then but here this was only measured then. Please reformulate ('at the end of the experiment in 9th generation measured...')

Line 21: It would be interesting to know which subgroups these are.

Line 25-26: This is not so clear from figures/data...

Lines 89-94: This is slightly unfair to say as clearly the random selection and high selection lines did not differ, only the time changed the seed yield. Would have been interesting to see what happened in soils without plants if microbes would have been continuously added like here and then at last stage plant grown.

For microbial communities it would be interesting to present also the (Bray-Curtis) distances from the time one (one-two, one-three, one-four...) and variation in them as a figure. This is said in the text but is not very clear from figure 2.

Presenting individual taxa with weak p-values derived from ANOVA is not necessary for the message of this paper and I would advice to think again if this is the main message wanted to convey here. Also, comparing only random and directional selection would make sense more than comparing to control.

Lines 154-157: Makes little sense, is this all unclassified bacteria together? What does this tell me actually?

For the network part it would be nice to see clearer representation of results so crucial for conclusions and even some statistics on it. Or is it all descriptive?

In the discussion it would be nice if data is compared more to other rhizosphere/soil studies and also acknowledge the other organisms important in soils. Comparing it to human gut makes some sense but is not very logical as there are papers on similar topics in soils.

Last, please have the pot-size in SI units

Reviewer #3 (Remarks to the Author):

In this study, the authors have tested and estimated the abundance of microbiome on the roots of *Brassica rapa*, and have attempted to correlate the results to the overall fitness of the plant. The authors claim that *B. rapa* selectively associates with a microbiome and enhances its seed biomass. Even though the question seeks attention; I believe that the experiments have not been rigorously tested. The results elaborated through the data analysis and the conclusions drawn from the figures and data does not directly complement each other. The overall writing has to be improved. Several sentences are confusing and needs to be restructured.

The fitness data from the phyllosphere biomass and seed count have not been directly correlated to the Nitrogen update or abundance.

Representative names of the three groups have to be improved. The use of the three conditions tested can be improved; Eg: Line 89: generation nine can be rewritten as 9th generation

Some of the terminologies used might be very confusing for the readers.

Lot of p values have been laid out without proper conclusions drawn against the overall experiment.

What do the authors mean by selection pressure?

In the methodology section, I think the methods can only be partially reproduced. The microbiome sequencing section does not have enough details to repeat the data analysis.

Figures can be made more informative. Eg: In figure 2, the generations(1-9) can be depicted inside the figure for the ease of readers. The legends for figures are poorly written.

In Figure 3, I did not understand PCoA of 16S rRNA gene OTUs? If I understand correctly, 16S rRNA genes were utilized to identify OTUs. Also, PCoA requires rigorous statistical tests (PERMANOVA) and this has to be mentioned in the figure as well as in the legends.

In figure 4, the network analysis carried out needs more refinement and also requires much more details. The circles at the perimeters have various sizes which need to be elaborated. I could not take adequate conclusions from this Figure.

More taxonomic details are required to explain the different OTUs found to be significant between the different conditions.

There seems to be quite a lot of variation across the 9 generations in the three conditions tested.

The percentage calculations for the relative abundance found across the conditions is incorrect

The statistical tools used are not ideal for drawing several of the conclusions and several of them have not been described well in the text and figures.

We thank the reviewers for their time in providing careful comments and suggestions. We performed additional analyses and revised figures and text in the main paper and supplementary materials. We believe the revised manuscript is much improved after incorporating the revision suggestions. The original reviewer comments are listed with author responses followed in bold font.

Reviewer 1

Reviewer 1: Line 74-76 “No hypotheses or expectations for potential patterns in biomass and seed yield? Confusing as written....is there a hypothesis or not?”

Response: We edited this sentence to state the hypothesis more clearly as follows in Lines 92-97: *“We hypothesized that selection pressure on rhizosphere microbiomes for increased biomass of a non-mycorrhizal plant host growing under nitrogen-limiting conditions will lead to the assembly of distinct, highly connected and interactive rhizosphere bacterial groups associated with plant host nitrogen uptake strategies. Additionally, we hypothesize that enhanced soil N cycling would be a key mechanism by which the rhizosphere microbiome of a non-mycorrhizal plant host would enhance biomass growth and seed yield.”*

Reviewer 1: “Reviewer 1: “Line 80: I don’t think is the place to outline the importance of these 3 treatments, but this needs to come out early in the results section so the reader can understand the results in the context and reason for each of the specific selection line and control treatments. Especially the random and control treatment? What do these treatments provide? The entire study hinges on these treatments and the reader needs to understand this upfront.

Line 88 and throughout Results: I feel it is necessary to move some brief methodology into each Results section. The “results” first style requires some well-placed text to provide context for the results being presented. Especially some of the experimental design, descriptions of selection lines, etc.

Line 92: Is the control treatment just the original field-collected and prepped inocula? Might want to clarify these types of delineations to help the reader with the main results.”

Response: We added a short summary (Lines 99-116) of the experimental design at the end of the introduction section to help readers better understand the control comparisons used to assess plant yield and NUE changes through selection pressure. All three treatments started with the same soil inoculum at the 1st generation. Additionally, the inclusion of figure 1 will help clarify the experimental design before the results section.

Reviewer 1: “Lines 94-100: Here is the initial result showing high biomass phenotypes don’t differ from those in random selection lines. Likewise, Table 1 shows the lack of differences in the measured plant phenotypes, aside from the isotope ratio.”

Response: We revised the text in the Results and Discussion sections (Line 120-126 and 283-289) to clarify that the final (9th) generation is the only generation that included seed yield. While the high biomass and random selection lines were not significantly different, the control line that included cryopreserved microbiomes (control against adaptation) showed clear and significant differences in yield from plants undergoing selection pressure of the microbiome, but no distinctions with randomly selected plants.

Reviewer 1: “Lines 103-104: It is hard to determine here if “selection line” and “treatment” are the same thing or something different.....needs to be clarified or rewritten to improve understanding.”

Response: Treatment and selection line were used interchangeably in the draft reviewed previously. We can see how this would be confusing for readers and have replaced “selection line” with “selection treatment” throughout the manuscript.

Reviewer 1: “Lines 120-122: Is the main takeaway here that no differences were determined after the first-generation harvest? Which should all be the same inocula, and you shouldn’t expect differences after generation 1? Or that through 9 generations of the selection lines and control treatments we finally start to see divergence of the selection line treatments from the control treatment in terms of community-level selection of the microbiome? Clarifying the text about expectations in the introduction and the results could be very helpful.”

Response: The reviewer has interpreted the ordination as we intended. We understand that we should highlight these main points with greater clarity in the results and discussion sections. The revised text are found in lines 181-187 and 282-289.

Reviewer 1: “Lines 136-157: Are the described patterns determined from comparing generations 1 and 8? Or across all generations? How was this done? Also, there should be some reference to the supplemental figure of the stacked bar plots of relative abundances?”

Response: For this analysis, we ran a model that assessed the effect of Generation, Selection treatment, and the interaction between the two on microbial abundances. The p-values and patterns reported are based on a posthoc test to compare relative abundances of the different microbial taxa in the three selection treatments from the 8th generation. We have included additional text to clarify this for the readers in the Supplementary Figure 2 legend and as it is cited in the text.

Reviewer 1: “Line 159: Since there is microbial data for each generation, why not determine community turnover through time?”

Response: We appreciate this excellent suggestion. Typically microbial turnover is for spatial data and not temporal data, so we were not able to use traditional turnover methods, such as betadiversity. We worked closely with our statistical consultant to look into similar methods for temporal data. We attempted to use the temporal betadiversity method outlined in Legendre 2018, but realized that our experimental design is not appropriate for this method or other temporal turnover methods. Analysis of temporal turnover methods require paired sites, which we didn’t have in this experiment. This would definitely be an interesting idea for future work, and perhaps other microbial ecologists that read our paper might pursue this idea.

Reviewer 1: “Lines 166-211: Very cool work here.”

Response: Thanks so much. Inspiration for the work is based partly on:

Li, L., Li, W., Zou, Q., & Ma, Z. S. (2020). Network analysis of the hot spring microbiome sketches out possible niche differentiations among ecological guilds. *Ecological Modelling*, 431, 109147.

Reviewer 1: “Lines 257-264: This paragraph feels out of place. The last two sentences specifically don’t relate to findings from the study and seem misleading. Consider reworking this paragraph to make the intent clearer.”

Response: We agree with the reviewer and have deleted this paragraph.

Reviewer 1: “Lines 272-277: These lines are confusing and don’t make a concise point? What about your ordination and analysis are you alluding to? What does a decrease in bacterial diversity really tell us?”

Response: We have edited the discussion to explain the potential implications of changes in bacterial diversity in the context of this experiment. We describe that the decline in bacterial diversity is an indication that ecological filtering may have occurred, where rare or less-relevant taxa are excluded over time (Lines 221-222).

Reviewer 1: “Lines 304-312: These sentences are incredibly speculative and do more harm than good. Overall, a large portion of the discussion feels speculative and not reinforced with specific results from the study.”

Response: We understand the reviewer’s concerns and removed this section from the discussion. Our original intent was to acknowledge that other significant functions of the microbiome could have changed (other than plant NUE-associated changes) but we simply did not measure these.

Reviewer 1: “Line 445: “fungal”? Don’t think fungi were characterized or analyzed in this study. But since it was left in there....why weren’t fungi included in the study? They obviously have a direct influence on plant productivity and resource use.”

Response: We removed the term ‘fungal’ from the text. We did not include fungal analysis in this study due to the rationale of the experiment. The rationale of studying Brassica plants is to examine how this specific functional group of plants may have evolved coordination strategies with soil bacteria to overcome soil nutrient limitations. Most terrestrial plants form associations with mycorrhizal fungi, but Brassica species characteristically do not. The AMF are essential in enhancing nutrient acquisition for their plant hosts, but some plant lineages lost this symbiotic relationship over evolutionary time. The network analysis performed in this rapid evolution study was helpful to provide some insights into potential coordination activities across bacterial taxa developed through the generations of selection that resulted in NUE and seed yield changes. We hope the results will inspire researchers to consider examining if bacterial group dynamics change in similar patterns with other Brassica species and if they contrast with other plant functional groups that form AMF associations. We added this explanation in Lines 89-92.

Reviewer 1: “Lines 458 – 465? I know these plots are in the supplement, but where are they referenced in the main manuscript?”

Response: We mention these plots in the results section under the subsection “Analysis of the rhizosphere bacterial community.”

Reviewer 2

Reviewer 2: “First of all, the title is not very informative on the content. Actually, I would not probably read article with this title. To me, more important findings are the selection of microbiomes for plant performance (and maybe a bit the network aspect). This could be better reflected in the title of the paper. (The sentence in lines 18-19 would reflect better what was done).”

Response: We have edited the title as the reviewer suggested: “*Selection pressure on the rhizosphere microbiome alters nitrogen use efficiency and seed yield in Brassica rapa.*”

Reviewer 2: “Also, only bacteria that were even filtered were used and studied in the experiment, how would conclusions change if more realistic systems including other microbes and even larger organisms would be included? Microbiome is not only bacteria and hence it is preferred to use exact terms also here.”

Response: As suggested by the reviewer, we now specify “bacterial community” when referring to data analyzed in the experiment on 16S rRNA gene sequences. We use the term ‘microbiome’ when describing the experimental design and the focus of the selection pressure. Although we designed the experiment to favor the passage of soil bacteria across generations of plantings, we could not eliminate the possibility that some fungi and viruses passaged through as well.

Reviewer 2: “It is also shame that the experiment was not longer and that seed biomass could not be measured earlier as now it is not clear when it was changed.”

Response: We agree with the reviewer that seed biomass measurements would be ideal throughout the generations of plantings. However, the time required for developing seeds would have prolonged the experiment 5-6x longer. We grew the plants for 10 days which was the amount of time before flower stalks began to emerge. The complete lifecycle for this genotype is approximately 50-60 days. The microbiome collection required destructive harvesting, which would have resulted in capturing microbial information that represents the end of the lifecycle instead of the active vegetative phase. Seed development of the *B. rapa* plants also required the soil to be partially desiccated at the end of the lifecycle to promote seed ripening, which alters the soil moisture environment at the time of collecting rhizosphere soil. We added Lines 123-126.

Reviewer 2: “Abstract line 19-20 this sounds like the 9th generation is the key and this would only happen then but here this was only measured then. Please reformulate (‘at the end of the experiment in 9th generation measured...’)”

Response: We edited the wording in this section as the reviewer suggested.

Reviewer 2: “It would be interesting to know which subgroups these are.”

Response: We included information that details bacterial phyla in Figure 4. Also, we edited this figure to include statistical measurements (PERMANOVA) that we originally placed in the supplementary materials section to make the results clearer.

Reviewer 2: “Line 25-26: This is not so clear from figures/data...”

Response: We moved supplementary Table 1 and supplementary Figure 4 to the main text to quantitatively show that microbial networks in the high biomass selection treatment had more connected microbial communities compared to the random and control treatments.

Reviewer 2: “It would be useful to have a clear figure on how experiments were performed and this could shorten a bit the text on it.”

Response: We added Figure 1 to diagram our experimental design and think it will be very helpful to readers.

Reviewer 2: “Presenting individual taxa with weak p-values derived from ANOVA is not necessary for the message of this paper and I would advice to think again if this is the main message wanted to convey here. Also, comparing only random and directional selection would make sense more than comparing to control.”

Response: We felt including this information was important to show how individual taxa within the community shifted in response to selection, which we believe helps to explain our network results and potential functional shifts in the microbiome (as we explain here and in the discussion, some of these bacterial families contain important members that are involved with plant growth promoting/N cycling functions, and could help explain the observed phenotypic changes between selection treatments). The control treatment is important to include because it provides a reference point to compare this non-adaptive microbiome to the two sets of microbiomes that adapted under growth chamber growing conditions. The control microbiomes experienced one generation of planting whereas the other two treatment microbiomes were passaged through nine generations.

Reviewer 2: “Lines 154-157: Makes little sense, is this all unclassified bacteria together? What does this tell me actually?”

Response: We deleted this result since it was likely not informative as mentioned by the reviewer.

Reviewer 2: “Why was filtering for OTUs set to 1% in any sample? This will include only very dominant species when diversity is high.”

Response: The removal of low abundance OTUs is a standard method used in microbiome sequence data analysis to reduce the noise introduced by poorly represented OTUs and increase the network robustness (Barberán et al 2012; Leite et al, 2018; Li et al, 2020). By focusing on the core microbiome, it also reduces the network complexity. We performed the network analysis on OTUs which are above 1% relative abundance in the samples. The soil environment in particular includes thousands of OTUs, of which the majority are rare and in very low abundance.

Barberán, A., Bates, S. T., Casamayor, E. O., & Fierer, N. (2012). Using network analysis to explore co-occurrence patterns in soil microbial communities. *The ISME journal*, 6(2), 343-351.

Leite, D. C., Salles, J. F., Calderon, E. N., Castro, C. B., Bianchini, A., Marques, J. A., ... & Peixoto, R. S. (2018). Coral bacterial-core abundance and network complexity as proxies for anthropogenic pollution. *Frontiers in microbiology*, 9, 833.

Reviewer 2: “In the discussion it would be nice if data is compared more to other rhizosphere/soil studies and also acknowledge the other organisms important in soils. Comparing it to human gut makes some sense but is not very logical as there are papers on similar topics in soils.”

Response: We have included examples from soils and removed the citations for human microbiome work, and also acknowledged other soil microorganisms in the conclusion. We think this helps keep the paper more on topic.

Reviewer 2: “It would have been also interesting to look at if selecting for the other extreme (small plants) would lead to similar network pattern compared to random. This cannot be changed of course but can be briefly discussed.”

“Also, only bacteria that were even filtered were used and studied in the experiment, how would conclusions change if more realistic systems including other microbes and even larger organisms would be included?”

Response: We have briefly addressed how future experiments could address these points at the conclusion of the discussion. Perhaps readers may get ideas for future experiments and move the field forward.

Reviewer 2: “please have the pot-size in SI units”

Response: The pot size is now reported in SI units.

Reviewer 2: “You used the average number of the 5 plants? And only above-ground biomass was measured?”

Response: We have clarified the methods, which was suggested by another reviewer. The biomass recorded was the total aboveground biomass for each pot and not an average. Roots were not included in the measurement.

Reviewer 2: “How was the random sample determined? i.e. the random sample based on randomness could be theoretically the same samples as high-performers. This would be interesting to see and discuss (also in terms of calculated variance).

One would expect the variation to become larger in random selection across soils but this is not looked at (in microbiomes or plants) but might be interesting.”

Response: We now include details of the random selection process in multiple sections of the paper and specify the use of a random number generator in Lines 439-440. Another reviewer suggested an earlier explanation of the selection process would help readers understand the experiment better.

We thank the reviewer for also suggesting to look at calculated variances of the random treatment. We did review the microbial and plant phenotypic variances between the selection treatments and found that they were largely equal between all the different treatments.

Reviewer 2: “The statistical models are quite simple (ANOVA) and one should note that most bacterial taxa are not distributed in the way that would allow these analyses. Please consider checking the assumptions (and variances) before presenting information on these groups and use correct models if assumptions are violated.”

Response: We have verified that the microbial abundances and all other data do meet the assumptions of ANOVA. Where assumptions were violated, we performed data transformations and included this in the paper.

Reviewer 2: “The heatmap creation should be specified and it could be noted that they are not overly informative here.”

Response: We have added more details about our heatmap creation in Lines 536-542.

Reviewer 2: “In Fig. 1 I would prefer to see boxplots (for variation and sample number) and colors would be also more pleasant.”

Response: We changed the barplots to boxplots and added colors as suggested.

Reviewer 2: “In Figure 2 the colors should be in legend and markers bigger. Colors could be also a scheme from light to red to see clearly the generation differences.”

“For microbial communities it would be interesting to present also the (Bray-Curtis) distances from the time one (one-two, one-three, one-four...) and variation in them as a figure. This is said in the text but is not very clear from figure 2.”

Response: We did try creating a color scale for this figure in red as suggested, but it was difficult to distinguish the generations from one another. We revised the legend as suggested. Additionally, while we appreciate the suggestion to make separate Bray-Curtis plots, we felt including seven additional ordinations might overwhelm the reader and detract from the main points of the manuscript.

Reviewer 2: “Fig. 4 (the key finding): it is not very clear from this figure that there are more connections in high biomass systems (main message). I would say that random looks very similar to the high biomass systems. Please add stats/clearer representation of the data. Supplementary fig 4 is much clearer but indeed does not show all data.”

Response: We agree that the first two figures alone may have been confusing for the reader. To show quantitatively that there are indeed differences in bacterial community density and rewiring in the high biomass line, we added figures 6C and 6D (formerly SF4 and Supplementary Table 1). In Figure 6C, we show the largest clusters for each selection line. The size of each node is proportional to the number of connections it has, and it is more apparent that taxa in the high biomass line indeed have larger connectivity than the random and control lines. Results from the NetShift analysis (Figure 6D) show that the high biomass treatment has greater density, total edges, exclusive edges, and lower average pathlength compared to the other two selection treatments. These results ultimately suggest more connections within the high biomass bacterial community.

As for the eLSA itself (ie figure 6A and 6B), each network analysis is built using all the samples in the corresponding group (~60 in FC, R, and HB). Therefore for the network characteristics (edge number, modularity, etc), we could not provide any statistics. Furthermore, eLSA compared to the traditional LSA already considers replicates in the calculation, which makes it more suitable for microbiome datasets (Xia et al, 2011).

We think these edits help make the results clearer.

Xia, L. C., Steele, J. A., Cram, J. A., Cardon, Z. G., Simmons, S. L., Vallino, J. J., ... & Sun, F. (2011, December). Extended local similarity analysis (eLSA) of microbial community and other time series data with replicates. In *BMC systems biology*(Vol. 5, No. 2, pp. 1-12). BioMed Central.

Reviewer 3

Reviewer 3: “Some of the terminologies used might be very confusing for the readers.”

Response: We added details in more sections that clarify terminologies. For example, the end of the introduction now includes a detailed summary of the selection pressures used in this experiment and their context (Lines 99-116).

Reviewer 3: “What do the authors mean by selection pressure?”

Response: In this experiment, selection pressure refers to the selection of a phenotype and the process we used to select for it. We selected for a plant phenotype (increased aboveground biomass) but the pressure was placed on the rhizosphere microbiome and not on plant genetics based on their collective abilities to modify phenotypes of the host plant. Specifically, we selected for the highest aboveground biomass in four pots out of 15 replicated units and composited the rhizosphere microbiome to use as inoculants for the subsequent generation. For the random treatment, the four pots out of 15 replicates were chosen at random using a

random number generator. We now include components of these important, defining details in the abstract, introduction, materials and methods, and discussion sections and added a Figure 1 diagram to remind the readers the experimental design details. Additional text in Lines 99-126 and 273-282.

Reviewer 3: “Representative names of the three groups have to be improved. The use of the three conditions tested can be improved; Eg: Line 89: generation nine can be rewritten as 9th generation”

Response: We revised the paper to better define the treatments in multiple sections and help readers understand the represented names of the treatments. We revised the generation term to include “9th generation” instead of “generation nine.”

Reviewer 3: “Lot of p values have been laid out without proper conclusions drawn against the overall experiment.”

Response: Another reviewer suggested adding more methods and rationale to the subsections of our results section, which we think helps to contextualize the p values and relate our analyses back to the experiment objective. We also added short statements to interpret the p-values at the end of each subsection

Reviewer 3: “The fitness data from the phyllosphere biomass and seed count have not been directly correlated to the Nitrogen update or abundance.”

Response: This is an excellent suggestion. We regressed our nitrogen data to plant productivity in the 9th generation and found that total foliar nitrogen was positively correlated with seed yield and aboveground biomass production. These new results have been added to the paper in Lines 170-173 as follows: “To link our foliar N measurements to plant productivity, we performed Pearson correlations using data from the 9th generation of the experiment. Total foliar N was found to have a strong positive correlation with aboveground dry biomass production ($df=10$, $R=0.89$, $p=1e-04$) (Figure 3A) and seed yield ($df=10$, $R=0.75$, $p=0.0051$) (Figure 3B).”

Reviewer 3: “The percentage calculations for the relative abundance found across the conditions is incorrect”

Response: The percentage calculations for microbial relative abundances are now corrected.

Reviewer 3: More taxonomic details are required to explain the different OTUs found to be significant between the different conditions.

Response: We have included more taxonomic details about the families found to be different in this revised section.

Reviewer 3: “The statistical tools used are not ideal for drawing several of the conclusions and several of them have not been described well in the text and figures.”

Response: We used standard microbiome sequence analysis tools as described in the materials and methods sections with citations of the Brazillian Microbiome Project pipeline and Quantitative Insights into Microbial Ecology (QIIME). The network analysis tools are not standard for amplicon sequence analysis and were developed using methods based on published work on network data analysis. Additionally, we added the correlation results from the 9th generation to link plant nitrogen uptake to seed yield and biomass production as the reviewer suggested.

Reviewer 3: “In the methodology section, I think the methods can only be partially reproduced. The microbiome sequencing section does not have enough details to repeat the data analysis.”

Response: We added more details about the microbial data analysis in the materials and methods section in Lines 480-578 and 516-521.

Reviewer 3: “Figures can be made more informative. Eg: In figure 2, the generations(1-9) can be depicted inside the figure for the ease of readers. The legends for figures are poorly written.”

Response: We have included the Generations and Selection treatment in the key for the legend in this figure and have also edited the other plots and legends according to suggestions from other reviewers.

Reviewer 3: “In Figure 3, I did not understand PCoA of 16S rRNA gene OTUs? If I understand correctly, 16S rRNA genes were utilized to identify OTUs. Also, PCoA requires rigorous statistical tests (PERMANOVA) and this has to be mentioned in the figure as well as in the legends.”

Response: The reviewer is correct. We used the 16S rRNA gene sequences to identify bacterial OTUs based on the publicly available database. We performed the PERMANOVA and found significant effects of generation, selection treatment, and generation x selection treatment. We included those results in the figure, in the figure legend, and in the text for clarity: Lines 178-181 and figure 4.

Reviewer 3: “There seems to be quite a lot of variation across the 9 generations in the three conditions tested”

Response: Is the reviewer referring to variance as indicated by the Bray-Curtis ordination plots? Although only 34% of the variance is explained in PCoA 1 and 2, it is typical and in the range expected for microbiome sequence data from environmental samples.

Reviewer 3: “In figure 4, the network analysis carried out needs more refinement and also requires much more details. The circles at the perimeters have various sizes which need to be elaborated. I could not take adequate conclusions from this Figure.”

Response: We have moved figures and a table from the supplementary materials section to the main paper to better explain the network analysis (figure 6CD) and to help contextualize the network analysis. We have also included an explanation of the node sizes in the key. Size of the node is proportional to the degree of the node (the number of connections a node has).

Reviewers' comments:

Reviewer #1 (Remarks to the Author):

Following the initial review and revisions, I believe the authors have addressed my comments and the comments of the other reviewers appropriately. I accept the revised manuscript.

Reviewer #2 (Remarks to the Author):

The authors have answered well most of the comments from previous round of review and changed the manuscript accordingly. It is much clearer now and especially benefits from a clear explanation of the treatments. What is now noticeable is that in the abstract and elsewhere it is said that the selection treatment changes plant growth and seeds while actually it looks from the data that both random selection and high-biomass selection have very similar effects on the plant growth and even bacterial communities (based on at least figure S2). This should still be made clearer: any selection by plant is changing the microbiome and affecting the plant while the selection for high biomass only affects the ^{15}N signatures. This itself is interesting though. However, the abstract and discussion should be adjusted accordingly + in the results section it should be clear what is the effect of the control soil without any selection of the plant and what is actually selected by presence of a plant AND by the selection of high biomass varieties. The change in the network configuration is one of the main effects but this is hard to quantify statistically and it would have really been nice to have multiple selection lines (random, low biomass...) to see the background of this selection.

The discussion is pretty repetitive and few of the paragraphs say the same thing 'bacteria changed and this potentially is related to the elevated N in leaves' – yet these two are not well linked in the results section. The observation that the random selection also changed the microbiomes considerably compared to at least control and that it also differed little from the high-biomass selection is kind of omitted. This should be discussed here – and authors should admit that the two controls used measure very different phenomena (one selection by presence of plants and the other one is stagnant situation without host plant). I do like that the negative feedbacks (and absence of them possibly in the high-biomass treatments) are discussed. The addition of call to study also other organism (and could add also under different plants, soils and nutrient conditions...) is good.

Overall, I still think the quality of the research is done to a high standard and article is well written. I want to thank authors also for changing the bar plots to boxplots where variability is shown better. My main concern is as stated above that the discussion and conclusions (that high-biomass differs considerably from also random treatment) are not fully based on the results presented here. This should not prevent the publication of this article but I do want to urge the authors to think how to present the effects related to the random selection better.

Very minor comment: it would be good to know which MiSeq was used (PE250 or PE300)

Hope you find the comments useful

Kind regards,
Emilia Hannula

Reviewer #3 (Remarks to the Author):

The authors have answered the questions raised and have improved the manuscript. I have no more questions.

Response to Reviewers

We thank the reviewers for their careful attention to the revision of the manuscript. We provided our responses below in **bold** following the original reviewer comments.

Reviewer 1:

Following the initial review and revisions, I believe the authors have addressed my comments and the comments of the other reviewers appropriately. I accept the revised manuscript.

Response:

We are thankful for your help in improving the paper.

Reviewer 2:

The authors have answered well most of the comments from previous round of review and changed the manuscript accordingly. It is much clearer now and especially benefits from a clear explanation of the treatments. What is now noticeable is that in the abstract and elsewhere it is said that the selection treatment changes plant growth and seeds while actually it looks from the data that both random selection and high-biomass selection have very similar effects on the plant growth and even bacterial communities (based on at least figure S2). This should still be made clearer: any selection by plant is changing the microbiome and affecting the plant while the selection for high biomass only affects the ¹⁵N signatures. This itself is interesting though. However, the abstract and discussion should be adjusted accordingly + in the results section it should be clear what is the effect of the control soil without any selection of the plant and what is actually selected by presence of a plant AND by the selection of high biomass varieties. The change in the network configuration is one of the main effects but this is hard to quantify statistically and it would have really been nice to have multiple selection lines (random, low biomass...) to see the background of this selection.

Response:

Thank you for your comments and assistance with improving this paper. We have revised the abstract (lines 20-23) and the discussion (lines 284-288) to more clearly state that the differences observed in traits such as seed yield and nitrogen agronomic use efficiency reported were between the high biomass and control treatment while the random treatment doesn't differ between the two. We also acknowledge that random selection does appear to have effects on plant and soil responses as you mentioned (lines 373-376).

The discussion is pretty repetitive and few of the paragraphs say the same thing 'bacteria changed and this potentially is related to the elevated N in leaves' – yet these two are not well linked in the results section. The observation that the random selection also changed the microbiomes considerably compared to at least control and that it also differed little from the high-biomass selection is kind of omitted. This should be discussed here – and authors should admit that the two controls used measure very different phenomena (one selection by presence of plants and the other one is stagnant situation without host plant).

Response:

All three treatments (high biomass, random, and control) were comprised on plants and microbiomes. We did not have a non-plant treatment. The control treatment consisted of microbiomes from the start of the first generation (first planting) and was identical to all three treatments. “Stagnation” is not a term we used in this manuscript, but instead we referred to as the “non-adaptive microbiome.” We explained that the control measures against adaption to the conditions of the growth chamber facility environment. The random selection treatment was conducted with a small sampling number of 15 plant-soil units, which we explained in the manuscript may not be a large enough pool to distinguish high biomass factors underlying the selection conditions. While the high biomass and random selection treatments behaved similarly for aboveground biomass, there are key differences between the two treatments that are important to keep in mind. For example, in figure 6C, which shows microbial interaction networks between the three selection lines along with taxonomic information, it shows that the high biomass treatment has a denser interaction network compared to both the random and control treatments, which was also verified quantitatively using NetShift. The taxa included in these networks indicate that the high biomass network includes more members from bacterial phyla such as Proteobacteria and Bacteroidetes in comparison to the random treatment microbiomes. Thus, we believe that selection for high biomass may not only alter microbiome composition, but may also promote group-level microbial interactions. We believe this aspect of microbiome assembly may help explain the increase in seed yield and NUE in the high biomass treatment compared to the control considering that many microbially mediated mineralization, solubilization, and uptake processes in soils are dependent on complex microbe-microbe coordination. Random selection does appear to have effects on microbial interaction networks and community composition, but these effects did not translate into differences in NUE and seed yield compared to the control, as was observed in the high biomass selection treatment. As shared in our initial reviewer responses, we believe the differences between the high biomass and random selection treatments are likely to become more pronounced with iterative generations (e.g. 30 generations of selection) which could be pursued in future work. We believe these results will inspire other researchers to design empirical studies that link to theoretical modeling of plant-microbial interactions.

I do like that the negative feedbacks (and absence of them possibly in the high-biomass treatments) are discussed. The addition of call to study also other organism (and could add also under different plants, soils and nutrient conditions...) is good.

Overall, I still think the quality of the research is done to a high standard and article is well written. I want to thank authors also for changing the bar plots to boxplots where variability is shown better. My main concern is as stated above that the discussion and conclusions (that high-biomass differs considerably from also random treatment) are not fully based on the results presented here. This should not prevent the publication of this article but I do want to urge the authors to think how to present the effects related to the random selection better.

Very minor comment: it would be good to know which MiSeq was used (PE250 or PE300)

Response:

We have added that we used PE300 for our sequencing (line 476).

Reviewer 3:

The authors have answered the questions raised and have improved the manuscript. I have no more questions.

Response:

We are thankful for your help in improving the paper.

REVIEWERS' COMMENTS:

Reviewer #2 (Remarks to the Author):

The authors have addressed my remaining comments and I am happy to see this article published.

Response to Reviewers

We thank the reviewers for their careful attention to the revision of the manuscript. Reviewers 1 and 3 accepted and approved the revisions in the previous submission. We provided our responses below in **bold** following the second reviewer's comments.

Reviewer 2:

The authors have addressed my remaining comments and I am happy to see this article published.

Response:

We are thankful for your help in improving the paper.